# Coherence mapping to identify the intermediates of multi-channel dissociative ionization
Jacob Stamm[1], Sung Kwon[1], Shawn Sandhu [1], Jesse Sandhu [1], Benjamin G. Levine [2,3] & Marcos Dantus [1,4] ✉

Identifying the short-lived intermediates and reaction mechanisms of multi-channel radical cation fragmentation processes remains a current and important challenge to understanding and predicting mass spectra. We find that coherent oscillations in the femtosecond time-dependent yields of several product ions following ultrafast strong-field ionization represent spectroscopic signatures that elucidate their mechanism of formation and identify the intermediate(s) they originate from. Experiments on endo-dicyclopentadiene show that vibrational frequencies from various intermediates are mapped onto their resulting products. Aided by ab initio methods, we identify the vibrational modes of both the cleaved and intact molecular ion intermediates. These results confirm stepwise and concerted fragmentation pathways of the dicyclopentadiene ion. This study highlights the power of tracking the femtosecond dynamics of all product ions simultaneously and sheds further light onto one of the fundamental reaction mechanisms in mass spectrometry, the retro-Diels Alder reaction.

Mass spectrometry (MS) is a powerful analytical technique widely used in the life sciences due to its high accuracy, speed, and sensitivity. Unlike nuclear magnetic resonance, rotational, vibrational, and electronic spectroscopy, the fragment ion pattern in MS cannot be predicted accurately via ab initio methodologies[1]. Unless a compound is already in a curated database, identification is challenging given that there is a large disparity between the number of distinct entries in databases and the more than 68 million known compounds. For example, as of 2015, only 1.8% of the spectra from an untargeted metabolomic experiment could be identified[2]. This knowledge gap affects drug development efforts, because unknown metabolites resulting from novel drugs cannot be readily identified by MS. Recent approaches incorporating machine learning have shown promise, especially on identifying specific classes of compounds[3,4]. High-level ab initio molecular dynamics simulations have made outstanding progress in predicting the main pathways for molecular fragmentation following ionization of small molecules[1,5]. However, obtaining statistically significant fragmentation patterns would be cost prohibitive because it would require tens or hundreds-of-thousands of trajectories involving multiple different electronic states. Traditional approaches based on quasi-equilibrium theory (QET) assume that the internal energy following ionization is completely randomized throughout all the vibrational modes before fragmentation[6]. Despite

the relative successes of QET, many dissociation processes proceed via non-ergodic pathways before energy is completely randomized[7], complicating MS prediction. Ultrafast studies capable of simultaneously tracking the yield of all (tens or hundreds) product ions can provide insight into the fragmentation process[8]. In particular, femtosecond time-resolved studies can identify non-ergodic processes by directly determining the reaction times[9,10]. Here, we augment this capability by mapping coherent vibrational dynamics in the time-dependent product ion yields to short-lived states of the system that are no longer reagent but not yet product, a term that has been referred to as the "transition states" of the reaction[11]. We will refer to these transient species as short-lived (often sub-picosecond) intermediates. Mapping product ions to their originating short-lived intermediates helps elucidate mechanistic information about the multiple independent and competing fragmentation pathways taking place following dissociative ionization.

In Fig. 1, we schematically illustrate how frequency, phase, and timing information from time-resolved product yield curves obtained by disruptive probing can be used to unravel the multiple fragmentation pathways that take place during dissociative ionization, a method we call coherence mapping. We consider the fragmentation of a molecular radical cation $M^{+\bullet}$ via two intermediate states with different vibrational

[1]Department of Chemistry, Michigan State University, S Shaw Ln, East Lansing, MI 48824, USA. [2]Department of Chemistry, Stony Brook University, John S. Toll Drive, Stony Brook, NY 11794, USA. [3]Institute for Advanced Computational Science, Stony Brook University, IACS Building, Stony Brook, NY 11794, USA. [4]Department of Physics and Astronomy, Michigan State University, Wilson Rd, East Lansing, MI 48824, USA. ✉e-mail: dantus@msu.edu

**Fig. 1 | Illustration of the disruptive probing and coherence mapping methodologies to determine intermediates from product-ion yields. a** During disruptive probing, the pump pulse ionizes the sample molecule *ABC*, causing fragmentation. The probe pulse disrupts the fragmentation by providing enough energy to alter the dissociation pathway, leading to depletions in some product yields and enhancements in others. **b** An example of how coherence mapping uses the residual oscillations in the time-resolved ion yields to reveal which intermediate(s) a particular product originates from. In this case, $Int_1^+$ and $Int_2^+$ have different vibrational signatures, which get mapped onto their fragmentation products by the probe pulse.

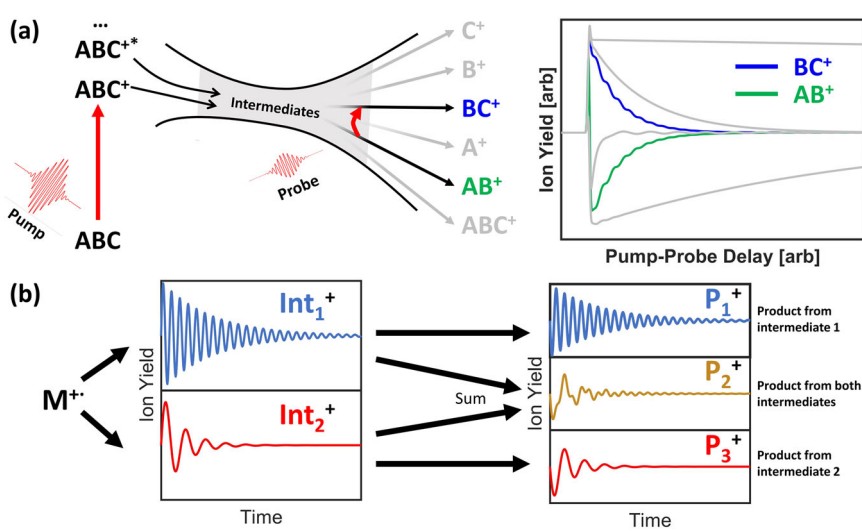

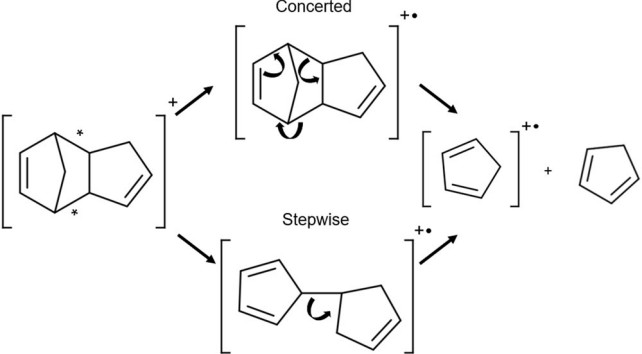

**Fig. 2 | The concerted and stepwise retro-Diels-Alder reaction for DCPD following ionization.** The retro-Diels-Alder scheme shows both the concerted and stepwise mechanism. The asterisks correspond to the two C-C bonds that break during the rDA reaction.

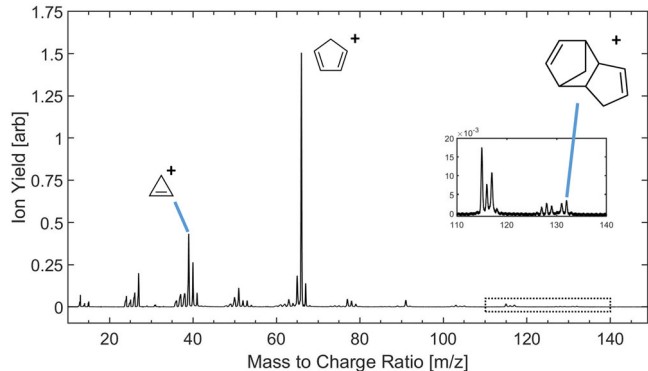

**Fig. 3 | Mass spectrum of DCPD obtained following SFI.** The spectrum was taken following irradiation with a single pump pulse of intensity $1.5 \times 10^{14}\,\text{W cm}^{-2}$. Several key ions are labeled with their most probable structure and the dashed region from m/z 110 to m/z 140 containing the molecular ion is magnified in the inset. Peaks attributable to $H_2O$, $O_2$, and $N_2$ have been removed for clarity.

signatures. Because the signatures get imprinted onto the product ion yields, their origin is revealed.

We chose to illustrate the power of coherence mapping by studying the ultrafast fragmentation of endo-dicyclopentadiene (DCPD) following strong-field ionization (SFI). Upon ionization, tens of different product ions

are observed, among them cyclopentadiene (CPD), which is produced by the retro-Diels Alder (rDA) reaction, an important mechanism in synthetic organic chemistry because it preserves stereospecificity[12,13]. The rDA reaction of a radical cation, shown in Fig. 2 for DCPD, was observed in one of the first studies addressing rDA reaction stereospecificity in MS[14]. The first ultrafast study of rDA reactions in neutrals was carried out by Zewail and coworkers on norbornene and norbornadiene[15]. Their study suggested both concerted and stepwise rDA mechanisms occurred. Our group analyzed the rDA reaction following strong-field ionization of cyclohexene, norbornene, and DCPD[16]. That study found that the mechanism in cyclohexane was ergodic in nature. For norbornene, coherent oscillations in the rDA yield suggested that energy randomization did not fully occur before product formation. However, the rDA reaction for DCPD was not analyzed in detail.

Here, we apply frequency and phase analysis to gain further insight into the fragmentation of DCPD following SFI. Using the time-resolved dynamics that lead to each ion, we determine the relative contributions of the concerted and stepwise mechanisms to the rDA reaction, as well as scrutinize the origin of the coherent oscillations observed in several key product ions. Aided by ab initio calculations, we identify the short-lived intermediates associated with the reaction pathways that give rise to the different product ions. This coherence mapping approach aids in the elucidation of fragmentation patterns that occur in MS and provides insight into the mechanistic details leading to the formation of multiple product ions.

## Results
### Fragmentation of DCPD
The experimental mass spectrum of endo-DCPD is shown in Fig. 3. It exhibits a small molecular ion peak (m/z 132), even at low intensity conditions. This indicates that, under the excitation and subsequent internal energy conditions of this experiment, the molecular ion preferentially fragments into the prominent peak at m/z 66. This suggests that the two bonds between the two cyclic rings of DCPD are prone to cleavage via a retro-Diels-Alder reaction following ionization (see Fig. 2). Additionally, a prominent peak at m/z 39 is observed, corresponding to $C_3H_3^+$, which is likely the cyclopropenium ion, formed by the elimination of acetylene from $CPD^+$[17]. Similar small-ringed fragments are found in the fragmentation pattern of toluene[17], suggesting that some of the other key fragments containing four, six, and seven carbons in the mass spectrum of DCPD may be cyclic, as in toluene.

### Time-resolved measurement of DCPD fragmentation
To study the fragmentation dynamics of DCPD, we employed disruptive probing[8] as described in the Methods section. Briefly, we used a strong

$(1.5 \times 10^{14}\ \text{W cm}^{-2})$ pump pulse to singly ionize the DCPD molecules, and a time-delayed weak $(3 \times 10^{13}\ \text{W cm}^{-2})$ probe pulse to track the fragmentation dynamics. Using this method, we tracked all fragment ion yields in the mass spectrum as a function of pump-probe delay to measure the molecular dynamics leading to each species.

The time-dependent yields for several key ions are shown in Fig. 4. We observe a spike in the yield of all ions at zero pump-probe delay, resulting from the addition of the pump and probe intensities when they are overlapped in time. The curves following this time-zero feature are

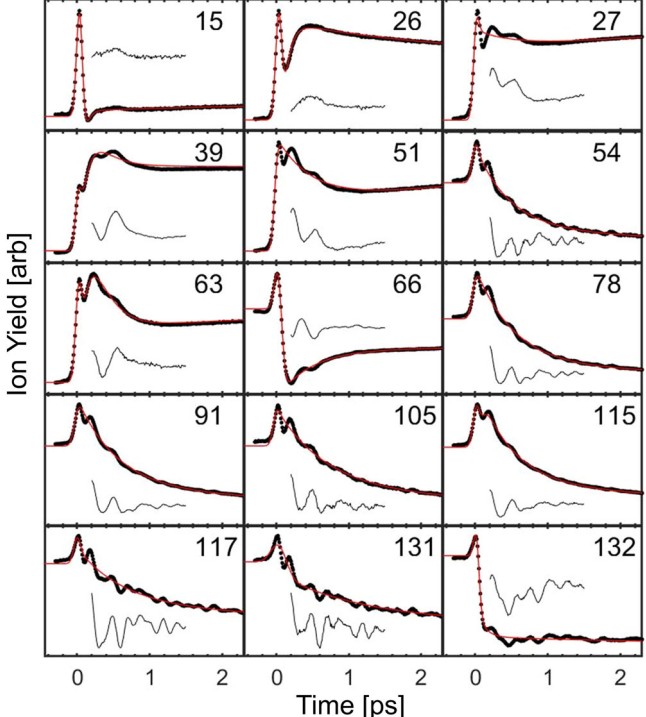

**Fig. 4 | Pump-probe delay dependent ion yields of several key fragments of DCPD.** Time-dependent ion yields are represented as black points, curve fits to Eq. (1) as red lines, and scaled residuals as offset black lines. Residuals, obtained by subtracting the data from the fit, have been scaled up by a factor of three. The spectrum was taken following irradiation with a pump pulse of intensity $1.5 \times 10^{14}\ \text{W cm}^{-2}$ and a weak probe pulse of intensity $3 \times 10^{13}\ \text{W cm}^{-2}$. All transients have been normalized such that negative times have yield of unity.

representative of the molecular dynamics leading to that particular ion. Generally, we observe that the yields of the larger fragment ions are depleted following time zero, in contrast to the smaller ions, whose yields are enhanced. This is due to the probe pulse promoting fragmentation of the larger ions as they form. In addition, we see coherent oscillations in the yields of several transients. If the coherent motion of the molecular ion modulated fragmentation, then one would expect all ions to display the same oscillations, as previously observed by our group in $CH_3NCS$[18]. However, this is not the case for the present molecule, where distinct frequencies are observed in different transients. To isolate the vibrational frequencies of these oscillations among different ions, we fit each time-resolved curve to the following function, derived in past works[8]:

$$P(t, \tau_1, \tau_2, \tau_3) = Ae^{-\frac{t^2}{s^2}} + \sum_{i=1}^{n} P_i(t, \tau_i) \tag{1}$$

where:

$$P_i(t, \tau_i) = A_i e^{-\frac{t}{\tau_i}} \left( 1 + erf\left( \frac{t}{s} - \frac{s}{2\tau_i} \right) \right) \tag{2}$$

In this equation, $A$'s are amplitude factors, $\tau$'s are exponential time constants, $t$ is a specific pump-probe delay time, $n$ is the number of exponential terms to fit to, and $s$ is a parameter related to the full width at half maximum ($\tau_{FWHM}$) of the pulse duration:

$$\tau_{FWHM} = 2\sqrt{\ln 2}\, s \tag{3}$$

### Frequency analysis of the pump-probe measurements of DCPD

Fitting the data in Fig. 4 to Eq. (1) allows us to extract oscillations from the residual by subtracting the fit from the data. While the Fourier transform is typically used for analysis of such oscillations, we instead use the maximum entropy method (MEM)[19] as shown in Fig. 5. This method was chosen because the MEM is better suited for analyzing the frequencies of short-lived oscillations like those shown in Fig. 4. A traditional Fourier transform exhibits a broad peak due to the strong damping of the oscillations. Note that the MEM spectra match the main features of the Fourier transform with much narrower peaks and with some minor peak shifting during retrieval. The multidimensional representation of the data indicates that the frequency at $90\ \text{cm}^{-1}$ dominantly modulates most of the fragments. Frequencies $135\ \text{cm}^{-1}$ and $195\ \text{cm}^{-1}$ are dominant in the larger fragments m/z ≥ 78 as well as m/z 54.

**Fig. 5 | Coherence mapping for the fragmentation of DCPD.** The top panel shows the sum of all maximum entropy method (MEM) spectra (black solid line), the sum of all Fourier transforms (black dotted line), and three specific ion MEM's. The bottom panel shows the MEM spectra of all ion residuals that show oscillations following subtraction by the fit obtained by Eq. (1). Each spectrum has been normalized by the maximum amplitude and labeled with a colored box indicating if its dominant frequencies come from the broken DCPD (red), or a mixture of broken and intact DCPD (green).

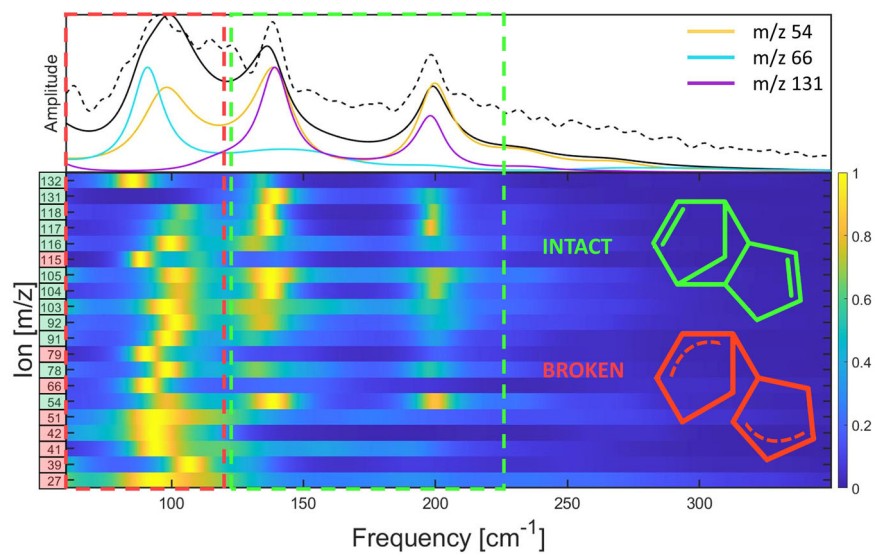

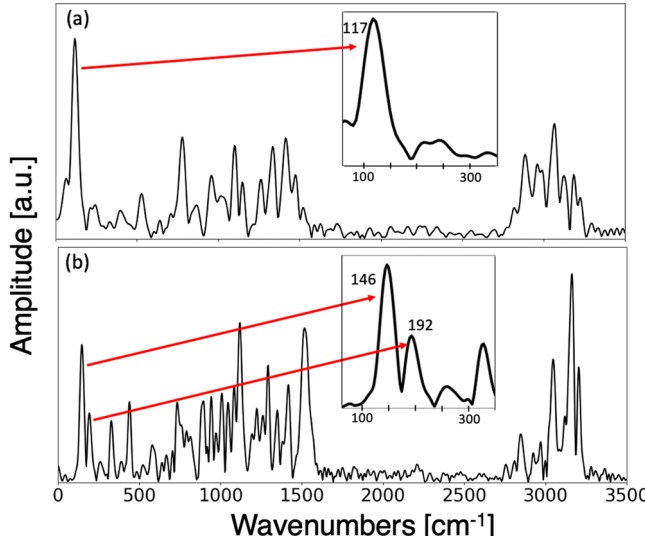

**Fig. 6 | Vibrational spectra obtained from the AIMD trajectories of singly-ionized DCPD.** Calculated vibrational spectra of the intermediate species for (**a**) stepwise rDA trajectories and (**b**) concerted rDA trajectories. Spectra were obtained by analysis of the AIMD trajectories (See Methods section). The insets show zoomed in vibrational spectra from 50 to 350 cm$^{-1}$.

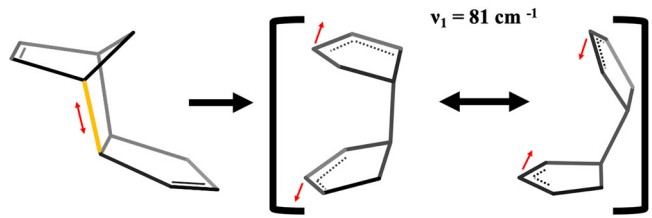

**Fig. 7 | Reaction progression leading to the activation of a vibrational mode of the broken bridge DCPD intermediate.** Reaction progression that shows activation of the $\nu_1$ vibrational mode that has a frequency of 81 cm$^{-1}$ calculated using uB3LYP/6-31++G**. One of the CPD bridging bonds in the DCPD radical cation is broken, allowing for the strain to be released along the axis of the bond cleavage. The red arrows illustrate the oscillations corresponding to this vibrational mode.

The vibrational frequencies of each fragment in Fig. 5 correspond to the spectrum of the intermediate species from which it originated, limited by the pulse width and sensitivity of specific vibrational motions to the probing process. The relative phase of the oscillations with respect to each other across different ions can be used to deduce the interplay among different channels. Thus, this information allows us to deduce the reaction mechanism for the dominant fragments, as discussed below.

### Ab initio molecular dynamics calculations

To better understand the origin of the experimentally observed coherent oscillations underlying the fragmentation of DCPD, we performed ab initio molecular dynamics calculations (AIMD). Further details of the calculations are presented in the Methods section. Surprisingly, we observed two distinct mechanisms for the rDA reaction from the single ionization of endo-DCPD: a stepwise and a concerted mechanism. We observed a stepwise mechanism in a majority of the trajectories (~90%). It should be noted that running exo-DCPD (instead of endo-DCPD) with the same method found only the stepwise mechanism for the rDA reaction. By taking the Fourier transform of the averaged-velocity autocorrelation of the concerted and stepwise rDA trajectories, we obtained a vibrational spectrum for each intermediate species (see Methods section for details). For each intermediate, frequencies corresponding to the C-H stretch (~3000 cm$^{-1}$), C=C stretch (~1500 cm$^{-1}$), and lower frequency bending modes (<1500 cm$^{-1}$) are observed in the

spectrum, showing activation of many modes following ionization. We focus specifically on the low frequency (0–350 cm$^{-1}$) region of these spectra given that the time resolution of our experiment limits the observation of higher frequencies. In this region, a mode appears at 117 ± 22 cm$^{-1}$ in the vibrational spectrum of the stepwise mechanism, while two different modes (146 ± 13 cm$^{-1}$ and 192 ± 12 cm$^{-1}$) appear in the vibrational spectrum of the concerted rDA mechanism (see Fig. 6 and Methods section). The errors in these frequencies were determined by using the standard deviation term obtained by fitting the peaks to a Gaussian. Note that these peaks are homogeneously broadened, with widths commensurate with those of the peaks from experimental measurements in Fig. 5.

As seen from the AIMD trajectories (See Supplementary Movie 1), the formation of CPD via a stepwise mechanism follows the consecutive cleavage of two bridge bonds (labeled with asterisks in Fig. 2). From an average of the trajectories, the first bond breaks within 260 fs, and the remaining bridge bond breaks in 2.25 ps. These timescales are reasonably commensurate with timescales obtained by fitting the m/z 66 transient (obtaining timescales of 485 ± 22 fs and 23 ps ± 3 ps). It is interesting to note that the π-electron rearrangement needed for dissociation is modulated by vibrational motion as seen previously in the McLafferty rearrangement[10]. However, in the concerted mechanism, the cleavage of both bridge bonds occurs nearly simultaneously after staying intact for a few picoseconds (See Supplementary Movie 2).

To help elucidate the origin of the oscillations seen in the time-resolved ion yields and calculated vibrational spectra, we performed electronic structure calculations to optimize intermediate structures found through AIMD and calculated the vibrational modes of said structures. A common intermediate structure that occurred during the rDA reaction is shown in Fig. 7. Additional minimized structures that have been identified are shown in Supplementary Fig. 3.

### Discussion

Coherence mapping analysis of the residual oscillations in the ion yields of several peaks following ionization of DCPD identified three important frequencies: 90, 135, and 195 cm$^{-1}$. These frequencies are not shared equally among all products, implying that they originate from different intermediates. The most dominant frequency lies at 90 cm$^{-1}$. By comparison to theory, we assign this to a large-amplitude motion of the rings of the intermediate species created upon cleavage of a single bridging bond (calculated to be 81 cm$^{-1}$ using uB3LYP/6-31++G**). This mode was also observed in the spectrum obtained from the velocity autocorrelation of AIMD trajectories (117 cm$^{-1}$, see Fig. 6). Ions with a yield modulated by this low frequency mode can be traced to the broken bridge intermediate which forms in 485 ± 22 fs (260 fs according to AIMD simulations). Experimentally, the damping time of the oscillations at 90 cm$^{-1}$ is 300 fs, which explains the breadth of the spectra shown in Fig. 5.

The higher frequency modes lie at 135 cm$^{-1}$ and 195 cm$^{-1}$. Comparison to theoretically derived frequencies finds close agreement with the two lowest vibrational modes of intact singly-ionized DCPD in its electronic ground state (calculated to be 150 cm$^{-1}$ and 201 cm$^{-1}$ using uB3LYP/6-31++G**). These modes were also observed in the spectrum obtained from the velocity autocorrelation of AIMD trajectories (146 and 192 cm$^{-1}$, see Fig. 6). These frequencies correspond to internal twisting modes of the intact DCPD and are observed in many of the larger fragment ions (m/z ≥ 78) as well as m/z 54. The long-lived oscillations imply that the molecular ion can remain intact for several picoseconds, which is also observed in the trajectories that showed a concerted rDA reaction (see Supplementary Movie 2). Many ions in Fig. 5 show oscillations from both the broken (90 cm$^{-1}$) and intact (135 and 195 cm$^{-1}$) intermediates. There are two possible explanations for this observation. The first is that both intermediates can subsequently fragment into almost any ion in the MS (given enough internal energy). The second possibility is that the fragment channels from each intermediate are mostly distinct, and it is the probe pulse that is causing the fragmentation of one intermediate into a channel of the other. We find it more probable that each product ion (with all 3 frequencies) forms

independently from each of the two mechanisms (i.e. each intermediate can lead to several of the product ions).

The pump-probe transient of the rDA product (m/z 66) from DCPD$^+$ exhibits mostly the low frequency mode at 90 cm$^{-1}$, which is attributed to the broken-bridge DCPD intermediate. This suggests that the rDA process in DCPD$^+$ dominantly occurs via a stepwise mechanism where the DCPD first breaks one of its bridging bonds, undergoes vibrational motion at 90 cm$^{-1}$ in this intermediate state, then breaks the second bridging bond to form the CPD$^+$ product (m/z 66). This matches the dominant pathway seen via AIMD simulations. In addition, experimentally we find small contributions from the 135 and 195 cm$^{-1}$ modes in the m/z 66 transient, indicating that a concerted mechanism from the intact molecular ion intermediate is also possible, although not as prevalent as the stepwise mechanism. This is supported by AIMD trajectories, which prevalently show the stepwise mechanism (~90%) as opposed to the concerted mechanism. Experimentally we can estimate the branching ratio between the stepwise and concerted mechanisms by comparing the amplitudes of the frequencies from the two different mechanisms. Doing this for the m/z 66 product ion we obtain a branching ratio of ~80% stepwise and ~20% concerted, in reasonable agreement with the AIMD simulations. Note that this estimate may be influenced by the relative sensitivity of the two different mechanisms to the probe pulse. If one of the pathways is more sensitive to probing than the other, then its measured ratio will be larger due to the more pronounced oscillations.

The second largest peak in the mass spectrum of DCPD corresponds to the cyclopropenium ion (m/z 39), which shows coherent oscillations at 90 cm$^{-1}$ (Fig. 5), similar to CPD$^+$ (m/z 66). Comparing the time-resolved curve of m/z 39 with m/z 66 reveals complementary trends in both the timescale and oscillations (see Fig. 4). The out-of-phase oscillations and complementary dynamics lead us to conclude that the broken bridge DCPD intermediate has two complementary pathways: one leading to m/z 66 and another leading to m/z 39 (among other fragments). Upon reaching this intermediate, it is prone to fragment into the dominant m/z 66 product, while the probe can excite this intermediate to a state that instead forms m/z 39. Oscillations at the 90 cm$^{-1}$ frequency modulate the preference for one pathway or another, leading to the out of phase oscillations and mirrored dynamics in the first picosecond of the m/z 39 and m/z 66 transients. It is worth noting that it is possible for m/z 39 to be formed from the fragmentation of m/z 66[17], as evidenced from the MS of neutral CPD[20]. We see evidence for this conversion, specifically in the long positive time delays (2 ps onwards) of the m/z 39 and m/z 66 transients (see Supplementary Fig. 2). A similar explanation applies to other fragments like m/z 27 and 51 i.e. they also result from a complementary pathway to m/z 66, a conclusion based on their mirrored dynamics and out of phase oscillations with m/z 66. Based on the results of Fig. 5, we can conclude that all these fragments originate from the broken-bridge intermediate due to their spectral amplitude at 90 cm$^{-1}$.

As for the product ions originating from the intact (135 cm$^{-1}$ and 195 cm$^{-1}$) DCPD$^+$ intermediate, most share similar dynamics in the pump-probe yields in Fig. 4 (m/z 54, 78, 91, 105, 115, 117, and 131) with a long (500 fs) depletion and generally long-lived oscillations. Apart from showing very similar dynamics, these fragments also share an equal branching ratio from the two DCPD$^+$ intermediates. In Fig. 5, we observe that these fragments show more spectral amplitude at frequencies corresponding to the intact intermediate relative to the other fragments. These similarities in both dynamics and spectral content indicate that fragments in this category may display their dynamics from the same precursor state, most likely the intact DCPD$^+$ intermediate. It should be noted that several more transients of this type are shown in the Supplementary Fig. 1.

## Conclusion

We use disruptive probing and coherence mapping to identify short-lived intermediates leading to multiple product ions and elucidate the fragmentation mechanism of endo-DCPD following strong-field ionization. The frequencies seen in the product ion transients and their respective phases inform one about their formation mechanism and what intermediates they

come from. Oscillations with frequencies of 90 cm$^{-1}$, 135 cm$^{-1}$, and 195 cm$^{-1}$ are identified, with different amplitudes for all product ions. Using ab initio molecular dynamics simulations and quantum chemical calculations, these frequencies were identified to be the lowest energy vibrational mode of the broken molecular ion (90 cm$^{-1}$), and the two lowest energy modes of the intact molecular ion (135 and 195 cm$^{-1}$). From these results, we conclude that the retro-Diels Alder reaction in endo-DCPD$^+$ is dominated by the stepwise mechanism, with small contributions from the concerted process. This study shows the power of being able to track the dynamics of all products simultaneously and sheds further light onto the fundamental reaction mechanisms responsible for the ion-patterns used for compound identification in mass spectrometry.

## Methods
### Experimental
We used ultrafast disruptive probing to measure the time of formation of all fragments after SFI[8]. The ionization source was a 35-fs Ti:Sapphire regeneratively amplified laser with a central wavelength of 800 nm and 1 kHz repetition rate. The pulses were compressed via a Multiphoton Intrapulse Interference Phase Scan, which measures spectral phase distortion for all orders and corrects them using a pulse shaper (MIIPS-HD, Biophotonics Solutions Inc) with a spatial light modulator (640 × 800, Hamamatsu)[21]. For all time-resolved measurements, the laser pulses were split with a beamsplitter, with a pump pulse intensity of $1.5 \times 10^{14}$ W cm$^{-2}$ and probe pulse intensity of $3 \times 10^{13}$ W cm$^{-2}$, calibrated by measuring the ratio between doubly and singly ionized argon ions[22]. The pulses were focused inside a Wiley-McLaren time-of-flight (TOF) mass spectrometer by a concave gold-coated mirror (f = 300 mm) and reached the previously mentioned intensities at the focus. The delay of the probe pulse with respect to the pump was changed via a nanometer-precision translation stage. Based on the ionization potential (IP) for DCPD, ~8.79 eV[23], the Keldysh parameter was calculated to be ~0.7, which indicates that tunnel ionization is the main mechanism as the classical ionization time scale occurs faster than one optical cycle[24]. Dicyclopentadiene (Millipore Sigma ≥96% purity) was used without additional purification. Results from gas chromatography-mass spectrometry measurements showed that the reagent was 96% endo-DCPD and 4% exo-DCPD. The endo-DCPD was degassed and loaded into the TOF chamber as a room temperature effusive beam through a needle valve. When the needle valve was closed, the vacuum pressure quickly returned to the baseline pressure of $5 \times 10^{-8}$ Torr. The static pressure during the experiments was kept below $1 \times 10^{-6}$ Torr for all data taken. The ions formed in the focal volume were accelerated towards a set of chevron dual microchannel plate detectors (Photonis) by a +1080 V extractor plate and +2168 V repeller plate with a 1 cm separation. The TOF was configured to measure cations and the ion signals generated at the microchannel plate detectors were computerized by an oscilloscope (LeCroy WaveRunner 610Zi, 1 GHz).

### Ab initio calculations and simulations
All ab initio molecular dynamics (AIMD) of endo-DCPD were performed on the singly-ionized ground state using uB3LYP/6-31++G**[25,26] with Wigner initial conditions to approximate experimental conditions, assuming rapid nonadiabatic relaxation within the manifold of cationic states. Trajectories were calculated using the GPU-accelerated quantum chemistry software TeraChem[27–29]. All trajectories were integrated up to 10 ps using the velocity Verlet integrator with a time step of 0.5 fs. Initial positions and momenta were sampled from the vibrational Wigner distribution using B3LYP/6-31++G** of the neutral ground state of endo-DCPD. All 26 trajectories were visualized using Avogadro[30,31] and were sorted into concerted or stepwise based on whether both bonds broke within the last 5% of the total reaction time. The intermediate structures of the endo-DCPD pathways identified via AIMD were optimized using uB3LYP/6-31++G**. To estimate the energetics of the products, the geometries and energies of CPD and CPD$^+$ were calculated separately using uB3LYP/6-31++G** and added together. For the coordinates of the optimized geometries

see Supplementary Data 1. The corresponding vibrations associated with the structures were calculated using the aforementioned level of theory.

To determine the effect of electron correlation on our results, we recalculated the energies of structures along the dissociation pathway obtained from uB3LYP using the high-level coupled cluster method, CR-CC(2,3)[32,33], with the basis set 6-31++G**. The CR-CC(2,3) calculations used restricted Hartree-Fock (for singlet ions and molecules) and restricted open-shell Hartree-Fock (for all non-singlet ions) as reference functions. The core orbitals (1s) on the carbons were kept frozen. We found excellent agreement in the relative adiabatic relaxation energies between uB3LYP and CR-CC(2,3), see Supplementary Fig. 3. All coupled-cluster calculations were performed using the GAMESS software package[34,35].

In order to identify the vibrational modes of the short-lived species that are populated following ionization, an averaged velocity autocorrelation of both the concerted and stepwise rDA trajectories was performed. For the concerted rDA trajectories, we used a 2150 fs autocorrelation window. For each trajectory, multiple starting points were used, beginning at 250 fs and ending at 300 fs in increments of 10 fs. For the stepwise rDA mechanism, 5 trajectories that had a reaction time greater than 1550 fs were selected. A 1250 fs window with multiple starting points beginning at 250 fs and ending at 300 fs in increments of 10 fs was used. For each mechanism, a Fourier transform of the averaged velocity autocorrelation was performed to obtain the vibrational spectrum. A Hamming window and zero-padding ($5\times$ number of points) was applied to the averaged velocity autocorrelation prior to calculating the Fourier transform.

## Data availability

The datasets generated during and/or analyzed during the current study are available from the corresponding author on reasonable request. Specific AIMD trajectories showing a stepwise and a concerted mechanism of the rDA reaction in DCPD is available as Supplementary Movies 1 and 2, respectively. Supplementary Data 1 contains the geometries of all optimized structures and an input file for TeraChem molecular dynamics used in this study. For more information about TeraChem inputs refer to the TeraChem manual.

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

## Acknowledgements

S.S. gratefully acknowledges Arnab Chakraborty for suggestions regarding coupled-cluster calculations. The authors gratefully acknowledge Dr. Steven Hurney for his assistance in the acquisition of the GC-MS data. This material is based upon work supported by the Air Force Office of Scientific Research under Award FA9550-21-1-0428. J. S. acknowledges funding from the U.S. Department of Energy, Office of Science, Office of Basic Energy Sciences, Atomic, Molecular, and Optical Sciences Program, under SISGER DE-SC0002325. This work used the SDSC Expanse GPU at San Diego Supercomputer Center through allocation CHE230046 from the Advanced Cyberinfrastructure Coordination Ecosystem: Services & Support (ACCESS) program, which is supported by National Science Foundation grants #2138259, #2138286, #2138307, #2137603, and #2138296. This work was supported in part through computational resources and services provided by the Institute for Cyber-Enabled Research at Michigan State University. BGL gratefully acknowledges support from the National Science Foundation CHE-1954519.

## Author contributions

Jacob Stamm contributed to Formal Analysis, Investigation, Writing (Original Draft), Writing (Review & Editing) and Visualization. Sung Kwon contributed to Investigation, Writing (Original Draft), Writing (Review & Editing), and Visualization. Shawn Sandhu contributed to Formal Analysis, Investigation, Writing (Original Draft), Writing (Review & Editing), and Visualization. Jesse Sandhu contributed to Validation and Visualization. Benjamin G. Levine contributed to Writing (Review & Editing), Supervision, and Funding Acquisition. Marcos Dantus contributed to Conceptualization, Methodology, Resources, Writing (Original Draft), Writing (Review & Editing), Supervision, Project Administration and Funding Acquisition.

## Competing interests

The authors declare no competing interests.
