## [Peer Review File · Communications Chemistry]

Reviewers' comments:

Reviewer #1 (Remarks to the Author):

The manuscript provides a beautifully executed follow up study of the Retro-diels- alder study in reference 14 and 15. In addition to the analysis offered in ref 15 for the observed coherent oscillations, here, the authors report that different products of the same parent ion exhibit different frequencies. With the advent of AIMD simulations, the authors attribute the different frequencies to two different mechanisms involving different vibrational oscillation of the transient species. This work is interesting and can be published after revisions addressing the following comments:

1. regarding fig. 6: For reliable comparison with the experimental data, please extend the zoomed in view to the same scale as figure 5. It will be clearer if there are additional peaks not appearing in the experimental data.
2. Also in fig. 6: The zoomed out view is not really discussed in the text. I am guessing that the feature around 3000 cm^{-1} can be attributed to C-H motion. Does the absence of features in the 1600-2800 region indicate that the single and double CC bonds are not vibrating? What is the motion associate with the 500-1500 cm^{-1} features?
3. On pg. 9 1st line states "As seen from the AIMD trajectories ". I have to say that it is not clear in the movie – specially since one has to carefully track a picosec long movie to identify a very abrupt bond breaking. (If the authors can emphasize in the movie the exact points, and maybe speedup the rest of the time – it will be very helpful) Also highlighting where and when the vibrational motion in fig 7 can be seen in the movie will be very insightful.
4. In the beginning of the discussion "Scissoring vibration" is used typically to describe the asymmetric vibration of of two groups bound to a single atom (e.g. in CH_2). The motion described in fig 7 is quite different so I'm not confident how the "scissoring" label helps to describe it. Please consider to revise.
5. Although the ultrafast dynamics broadens the frequencies (also nicely seen in the fig 6 theory, but not compared with experiment), the authors use very precise values, e.g. 81 cm^{-1} , and 117 cm^{-1} that are assigned to the 90 cm^{-1} seen in the experiment. Are the shifts significant within the errorbars? should they be considered and discussed? Also in the experiment different masses show shifts (e.g. the ~90 peaks of mass 115 and 116 are shifted) – are these shifts significant? Please revise the discussion.
6. On pg. 12, the authors suggest a 90% contribution from the stepwise mechanism. One should be careful, as different mechanisms can have very different sensitivities to the disruptive probe pulse.
7. Furthermore, did the authors calculate the dynamics of the exo-DCPD that has a 4% presence in their sample? (according to SI) that is comparable with the 10% contribution attributed to a concerted mechanism. Indeed, one should provide some evidence that the coherence frequencies allow disentangling different transient species of different mechanisms and not different isomers inside the sample (which can be by itself an interesting application of the method). It is important to clarify this point as in my opinion, the potential for broad interest in this work stems from the identification of competing mechanisms...
8. In section 4.1: Oscillations in mass 66 are not clearly seen in figure 4 (same for mass 39). I suggest a zoom in view to show the oscillations, such that it will be possible to see if they are out of phase.
9. Out of phase oscillation indicate competing products of the same intermediate. Assignment of mass 39 as secondary product of mass 66 can suggest in-phase oscillations as when 66 is disrupted - there will be less 39 as well. Unless the disruption occurs after 66 is already formed, but then the oscillation

frequency would not be associated with the undissociated parent ion. Although I can agree that 39 can be a secondary product of 66, one should revise the text to better explain the consistent logic behind the proposed assignment.

10. Same comments (8-9) apply to the other secondary dissociation product assignments in sections 4.2-4.5.

Minor comments:

11. In fig. 2, the Asterix indicating the bonds that break is not visible in the figure.

12. On pg. 8, 3 lines from bottom: transient state not necessarily transition-state. I assume it's a typo.

13. In several places there are typos in reference to figures I guess that appear as "(see ??)"

14. On pg. 14, please revise the tenses of the "Studies like this shows ..." sentence.

Reviewer #2 (Remarks to the Author):

The manuscript "Coherence Mapping to Identify the intermediates of Multi-Channel Dissociative Ionisation" by Stamm et al. details ultrafast pump-probe ion yield measurements on dicyclopentadiene (DCPD), focussing on the retro Diels-Alder reaction following ionisation. In particular the authors singly ionise DCPD with a strong-field pump pulse, and then modulate the fragmentation dynamics with a time-delayed probe pulse. This 'disruption' of the fragmentation processes manifests itself as depletion/enhancement of particular fragmentation channels, as well as coherent oscillations in the observed yield. The coherent oscillations in the time-domain are then analysed in terms of their frequency components, which can directly be correlated to the underlying vibrational structure of the intermediates from which a particular fragment originated. This allows the authors to observe a signature for a particular reaction pathway (here stepwise vs. Concerted), and with the aid of ab initial calculations assign each fragment to a particular channel.

The presented approach is a powerful and elegant way to gain insight into the (cation) fragmentation dynamics that are prevalent in MS of larger systems. This is clearly an important development, in particular with a view to predicting complex fragmentation patterns. The chosen process, a retro Diels-Alder reaction, is of high importance to chemistry, while the method of 'Coherence Mapping' should be fairly universally applicable. This is an important development and warrants publication in Commun. Chem. That said, the manuscript is currently missing important details that are needed in order to fully understand and reproduce the shown data. The discussion should furthermore be significantly extended, as further outline below. I therefore generally recommend publication after major revisions and further review.

Specifically:

-Most urgently, in Figure 5 the ion MEMs shown at the top (for m/z 54, 66, 131) do not correspond to the data shown in the bottom half. In particular for m/z 66 and 131. My feeling is that the color labels in the top figure are swapped!?

-In order to extract vibrational frequencies from the time-resolved data the dynamics component is removed by fitting this with (a sum of) exponential decays representing the lifetimes of intermediate

states. I fully agree with this approach, however the resulting fit is not shown in the manuscript (or SI), making it hard to judge how well this described the data. The extracted time-constants are furthermore not given. I strongly recommend that (at least for selected fragments) the actual raw data points, along with the fit and the extracted residuals are shown. The time constants should be discussed and compared to those available in the literature.

-The authors state that “While the Fourier transform is typically used for analysis of such oscillations, we instead use the maximum entropy method (MEM)” - could they comment on why they use MEM instead of the more common FT?

-At the top of page 8 the authors state that “The vibrational frequencies of each fragment in Figure 5 correspond to the vibrational spectrum of the intermediate species from which it originated”. I think this is somewhat overstated. The observed frequencies corresponds to frequencies of the intermediate, they do not represent the entire vibrational spectrum of it.

-For the performed AIMD simulations, how many trajectories were simulated? How were these classed into stepwise or concerted reaction mechanisms?

-I feel the discussion could be significantly extended:

--For the discussed correlations it is impossible to see from the small panels in Figure 4 if oscillations are in phase or not. I recommend inserting additional figures that shown in detail the dynamics traces for the particular fragments discussed (e.g. m/z 66 and 39), as well as the extracted residuals showing the oscillatory behaviour.

--The discussion should moreover be extended to further fragment channels. For example, m/z 79, which shows very different behaviour from m/z 78. Or m/z 115 which is by far the heaviest fragment that appears to follow the step-wise mechanism.

--The authors draw the qualitative conclusion that while the rDA process is accessible from both mechanisms, it is dominated by the stepwise reaction. Can they also draw more quantitative conclusions regarding the importance of both channels? Or what would be needed in order for more quantitative insight?

-The overall structure of the manuscript does not make for easy reading. Especially placing the “Methods” in between the “Results” and “Discussion” section is very unusual. I highly recommend placing this either directly after the “Introduction” or at the end of the manuscript. Furthermore, breaking the “Discussion” into small paragraphs each with their own subheading really breaks the flow of the text. I recommend extending the “Discussion” (as outlined above) and creating a single flowing text.

Some minor typos and corrections:

Figure 2 - asterisks highlighting C-C bonds are missing

Figure 6 uses different font and style to the other figures (e.g. round brackets for units instead of square ones)

Page 12 - several LaTeX references missing (“See ??”).

Several references are incomplete: 10, 15, 17, 25, 29, 32

Reviewer #3 (Remarks to the Author):

The authors present a study of strong-field ionization of the dicyclopentadiene molecule, in which they follow a retro-Diels-Alder reaction of the cation with pump-probe mass spectrometry with a disruptive probe pulse. They observe coherent oscillations in the final fragment ion yields that they attribute to vibrational movement of two distinct reaction intermediates belonging to two different reaction pathways (one concerted, one stepwise).

Such information is highly relevant for the study of cationic fragmentation processes and very challenging to come by with other methods. However, it seems challenging or even impossible with their experimental approach to strictly assign a final product to either of the intermediates, which would be an important second step. The results are novel and relevant for chemists and physicists working in mass spectrometry and photoionization studies. From this point of view, I think the study is suitable for publication in communications chemistry.

However, I have concerns regarding the comparison between theory and experiment and the lack of discussion of parts of the data. Some part of the discussion is difficult to follow because there are no graphs showing the details being discussed. If these concerns can be appropriately addressed, I will recommend publication.

1. Electronic state population of the molecular cation after tunnel ionization

The authors perform a tunnel-ionization experiment, but base their theoretical investigation on the assumption that only the cationic ground state is populated. Is there any argumentation why tunnel ionization should preferentially create ground state cations? Are there references supporting this? In their manuscript the authors state "This indicates that the molecular ion is metastable and preferentially fragments into the prominent peak at m/z 66,". I believe, the mass-selected VUV threshold photoelectron spectrum of dicyclopentadiene (published in [1]) indicates that the DCPD cation is stable in its electronic ground state (without significant vibrational energy). This would indicate that the observed fragment ions are produced from electronic excited states of the cation, populated in the tunnel ionization experiment. Consequently, the comparison to calculations for the cationic ground state would be questionable and would require additional justification. It may be possible that all these electronic states of the cation share the same vibrational signatures depending on whether one of the bridging bonds is broken or not, but this cannot just be assumed without argumentation.

[1.] Pan, Z., A. Puente-Urbina, A. Bodi, et al. *Chemical Science*, 2021. 12(9)

2. Why do most products show oscillations with frequencies of both intermediates?

The data in figure 5 shows that most product channels oscillate with all three identified vibrational frequencies. This can either be explained if any of these products is formed via both intermediates, or because the disruptive probe exchanges population between the fragmentation channels (One intermediate is depleted by the disruptive probe toward a product channel of the other intermediate). I think both possibilities need to be discussed. In general, more discussion of the data in the bottom panel of figure 5 would improve the paper.

3. Discussion of relative phases of oscillations

The authors discuss the relative phases of the oscillations and complementary trends observed in different product channels. These phases cannot be observed in the provided graphs. For this discussion it is necessary that the authors provide plots of the residuals showing these oscillations (at the very least in the SI), such that the reader can follow the discussion.

Minor points:

- The legend in figure 5 has masses 66 and 131 interchanged.
- The time scales given on page 9 do not agree with what I see from the video. The video seems to show bond breaking after 60 fs and after 6 ps rather than 260 fs and 2.25 ps.
- In the discussion the CPD+ ion is referred to as an intermediate. If I understand correctly, the out-of-phase oscillations of m/z 51 and 39 (not shown!) when compared with m/z 66 shows that these products compete with the m/z 66 (CPD+) product. I cannot see any argument why this means that m/z 66 needs to be an intermediate for the production of m/z 51 and 39. This needs to be explained more.
- A comment on the accuracy of the MEM method would be helpful. Are the differences in the 90 cm⁻¹ modes (position as well as widths) observed for the different fragments in figure 5 significant? Could these be indications of different electronic states, with slightly different vibrational frequencies leading to these products?
- Is the low number of trajectories calculated statistically sufficient? Only two or three trajectories seem to show the concerted decay, is this correct?

We thank the reviewers for their detailed comments on our paper. We have carefully revised our manuscript, marking our revisions in red type. Below (in blue) are our point-by-point responses to the helpful remarks and suggestions made by the reviewers, along with a description of the changes we have made in the manuscript and the supplementary material.

The Authors

Reviewers' comments:

Reviewer #1 (Remarks to the Author):

The manuscript provides a beautifully executed follow up study of the Retro-diels- alder study in reference 14 and 15. In addition to the analysis offered in ref 15 for the observed coherent oscillations, here, the authors report that different products of the same parent ion exhibit different frequencies. With the advent of AIMD simulations, the authors attribute the different frequencies to two different mechanisms involving different vibrational oscillation of the transient species. This work is interesting and can be published after revisions addressing the following comments:

1. regarding fig. 6: For reliable comparison with the experimental data, please extend the zoomed in view to the same scale as figure 5. It will be clearer if there are additional peaks not appearing in the experimental data.

Answer: We agree that there should be consistency between plot axes. The data in the inset of Figure 6 has been replotted to match the axes of Figure 5. See the updated figure below.

2. Also in fig. 6: The zoomed-out view is not really discussed in the text. I am guessing that the feature around 3000 cm⁻¹ can be attributed to C-H motion. Does the absence of features in the 1600-2800 region indicate that the single and double CC bonds are not vibrating? What is the motion associate with the 500-1500 cm⁻¹ features?

Answer: We agree that discussion of other peaks in the spectra is needed to give a clearer picture as to what is being measured. The features around 3000cm⁻¹ are indeed attributable to the C-H stretches. While

there are no frequencies present in the 1600-2800 cm^{-1} region, we attribute the C=C bond vibrational frequencies to the $\sim 1500\text{ cm}^{-1}$ region. The motions seen in the 500-1500 cm^{-1} region are attributable to the fingerprint region of the IR spectrum, mainly consisting of bending vibrations.

We have added some discussion regarding these peaks in the manuscript:

To better understand the origin of the experimentally observed coherent oscillations underlying the fragmentation of DCPD, we performed ab initio molecular dynamics calculations (AIMD). Further details of the calculations are presented in the Methods section. Surprisingly, we observed two distinct mechanisms for the rDA reaction from the single ionization of endo-DCPD: a stepwise and a concerted mechanism. We observed a stepwise mechanism in a majority of the trajectories ($\sim 90\%$). It should be noted that running exo-DCPD (instead of endo-DCPD) with the same method found only the stepwise mechanism for the rDA reaction. By taking the Fourier transform of the averaged-velocity autocorrelation of the concerted and stepwise rDA trajectories, we obtained a vibrational spectrum for each intermediate species (see Methods 5.2 section for details). For each intermediate, frequencies corresponding to the C-H stretch ($\sim 3000\text{ cm}^{-1}$), C=C stretch ($\sim 1500\text{ cm}^{-1}$), and lower frequency bending modes ($< 1500\text{ cm}^{-1}$) are seen in the spectrum, showing activation of many modes following ionization. We focus specifically on the low frequency (0-250 cm^{-1}) region of these spectra given that the time resolution of our experiment limits the observation of higher frequencies. In this region, a mode appears at $117 \pm 22\text{ cm}^{-1}$ in the vibrational spectrum of the stepwise mechanism, while two different modes ($146 \pm 13\text{ cm}^{-1}$ and $192 \pm 12\text{ cm}^{-1}$) appear in the vibrational spectrum of the concerted rDA mechanism (see Figure 6 and Methods section).

3. On pg. 9 1st line states “As seen from the AIMD trajectories “. I have to say that it is not clear in the movie – specially since one has to carefully track a picosec long movie to identify a very abrupt bond breaking. (If the authors can emphasize in the movie the exact points, and maybe speedup the rest of the time – it will be very helpful) Also highlighting where and when the vibrational motion in fig 7 can be seen in the movie will be very insightful.

Answer: We agree that the video in its present state makes it difficult to see the moment of bond cleavage as well as the vibrational motion mentioned in Figure 7. The Supplemental Video has been altered to better emphasize the moment of bond cleavage and to highlight the vibrational motion shown in Figure 7.

4. In the beginning of the discussion “Scissoring vibration” is used typically to describe the asymmetric vibration of two groups bound to a single atom (e.g. in CH_2). The motion described in fig 7 is quite different so I’m not confident how the “scissoring” label helps to describe it. Please consider to revise.

Answer: This is indeed a misuse of terminology. “the scissoring mode” has been changed to “a vibrational mode”. “a scissoring motion that modulates...” has been deleted.

5. Although the ultrafast dynamics broadens the frequencies (also nicely seen in the fig 6 theory, but not compared with experiment), the authors use very precise values, e.g. 81 cm^{-1} , and 117 cm^{-1} that are assigned to the 90 cm^{-1} seen in the experiment. Are the shifts significant within the errorbars? should they be considered and discussed? Also in the experiment different masses show shifts (e.g. the ~ 90 peaks of mass 115 and 116 are shifted) – are these shifts significant? Please revise the discussion.

Answer: We agree with the statements made by the reviewer. The certainty with which these values are known can help give a better interpretation of our measurements. The following changes have been implemented:

First, we highlight the broadening of the peaks seen both experimentally and theoretically.

Second, error bars have been reported when discussing the theoretical frequencies. The error bars were obtained by curve fitting the vibrational peaks to a gaussian and using the standard deviation term. The error bars for the frequencies of various mechanisms are as follows: Vibration 1 for concerted mechanism = $146 \pm 13\text{ cm}^{-1}$. Vibration 2 for concerted mechanism = $192 \pm 12\text{ cm}^{-1}$. Vibration 1 for stepwise mechanism = $117 \pm 22\text{ cm}^{-1}$. Finally, a comment about the shifts seen in Figure 5 is made to elaborate on whether the shifts are relevant or not.

The following changes have been made:

We focus specifically on the low frequency (0-250 cm^{-1}) region of these spectra given that the time resolution of our experiment limits the observation of higher frequencies. In this region, a mode appears at $117 \pm 22 \text{ cm}^{-1}$ in the vibrational spectrum of the stepwise mechanism, while two different modes ($146 \pm 13 \text{ cm}^{-1}$ and $192 \pm 12 \text{ cm}^{-1}$) appear in the vibrational spectrum of the concerted rDA mechanism (see Figure 6 and Methods section). The errors in these frequencies were determined by using the standard deviation term obtained by fitting the peaks to a Gaussian. Note that these peaks are homogeneously broadened, with widths commensurate with those of the peaks from experimental measurements in Figure 5.

Fitting the data in Figure 4 to Equation 1 allows us to extract oscillations from the residual, following subtraction of the fit from the data. While the Fourier transform is typically used for analysis of such oscillations, we instead use the maximum entropy method (MEM) as shown in Figure 5. This method was chosen because the MEM is better suited for analyzing the frequencies of short-lived oscillations like those shown in Figure 4. A traditional Fourier transform exhibits a broad peak due to the strong damping of the oscillations. Note that the MEM spectra match the main features of the Fourier transform with much narrower peaks and with some minor peak shifting during retrieval. The multidimensional representation of the data indicates that the frequency at 90 cm^{-1} dominantly modulates most of the fragments. Frequencies 135 cm^{-1} and 195 cm^{-1} are dominant in the larger fragments $m/z \geq 78$ as well as $m/z 54$.

6. On pg. 12, the authors suggest a 90% contribution from the stepwise mechanism. One should be careful, as different mechanisms can have very different sensitivities to the disruptive probe pulse.

Answer: While we agree with this statement as it relates to experimental data, the 90% contribution number mentioned in this case is derived from 26 AIMD trajectories in the “stepwise” or “concerted” cases.

7. Furthermore, did the authors calculate the dynamics of the exo-DCPD that has a 4% presence in their sample? (according to SI) that is comparable with the 10% contribution attributed to a concerted mechanism. Indeed, one should provide some evidence that the coherence frequencies allow disentangling different transient species of different mechanisms and not different isomers inside the sample (which can be by itself an interesting application of the method). It is important to clarify this point as in my opinion, the potential for broad interest in this work stems from the identification of competing mechanisms...

Answer: While it is a reasonable suspicion that the 10% concerted contribution may be from the exo-DCPD impurity, AIMD simulations on exo-DCPD find exclusively stepwise rDA reactions. Therefore, the 10% concerted in our sample cannot be attributed to exo-DCPD.

The following sentence has been added:

Further details of the calculations are presented in the Methods section. Surprisingly, we observed two distinct mechanisms for the rDA reaction from the single ionization of endo-DCPD: a stepwise and a concerted mechanism. We observed a stepwise mechanism in a majority of the trajectories (~90%). It should be noted that running exo-DCPD (instead of endo-DCPD) with the same method found only the stepwise mechanism for the rDA reaction. By taking the Fourier transform of the averaged-velocity autocorrelation of the concerted and stepwise rDA trajectories, we obtained a vibrational spectrum for each intermediate species (see Methods 5.2 section for details).

8. In section 4.1: Oscillations in mass 66 are not clearly seen in figure 4 (same for mass 39). I suggest a zoom in view to show the oscillations, such that it will be possible to see if they are out of phase.

Answer: This is true, as the comparison between the oscillations are important for the discussion section. The data in Figure 4 has been converted to raw data points, the fit used to obtain the residual oscillations is shown as a red line, and a black scaled residual line for all ions has been added above each transient to enable phase comparisons. (See the updated figure below).

9. Out of phase oscillation indicate competing products of the same intermediate. Assignment of mass 39 as secondary product of mass 66 can suggest in-phase oscillations as when 66 is disrupted - there will be less 39 as well. Unless the disruption occurs after 66 is already formed, but then the oscillation frequency would not be associated with the undissociated parent ion. Although I can agree that 39 can be a secondary product of 66, one should revise the text to better explain the consistent logic behind the proposed assignment.

Answer: The reviewer's observation is indeed critical to the interpretation of this experiment. We agree that an assignment of competing reaction channels explains the out of phase oscillations between mass 39 and mass 66 better than a secondary product explanation. While this agrees with the time-resolved data, it

disagrees with the literature on similar dynamics as well as the NIST spectrum of m/z 66 (all of which shows mass 39 can be a secondary product of 66). The best way to mesh these two facts is the following hypothesis/assignment: The out of phase dynamics between mass 39 and mass 66 is due to the probe disrupting an intermediate state (the broken bridge DCPD⁺) that would have made mass 66 into a state that instead makes mass 39 (as well as other products with mirrored dynamics to 66). The formation of 39 from 66 is also a possible pathway, and one that we see specifically at long timescales, a figure showing this has been added to the Supplementary Materials.

This argument has been explicitly discussed in the paper in the following way:

The second largest peak in the mass spectrum of DCPD corresponds to the cyclopropenium ion (m/z 39), which shows coherent oscillations at 90 cm^{-1} (Figure 5) similar to CPD⁺ (m/z 66). Comparing the time-resolved curve of m/z 39 with m/z 66 reveals complementary trends in both the timescale and oscillations (see Figure 4). The out-of-phase oscillations and complementary dynamics lead us to conclude that the broken bridge DCPD intermediate has two complementary pathways: one leading to m/z 66 and another leading to m/z 39 (among other fragments). Upon reaching this intermediate, it is prone to fragment into the dominant m/z 66 product, while the probe can excite this intermediate to a state that instead forms m/z 39. Oscillations at the 90 cm^{-1} frequency modulate the preference for one pathway or another, leading to the out of phase oscillations and mirrored dynamics in the first picosecond of the m/z 39 and m/z 66 transients. It is worth noting that it is possible for m/z 39 to be formed from the fragmentation of m/z 66, as evidenced from the MS of neutral CPD. We see evidence for this conversion, specifically in the long positive time delays (2 ps onwards) of the m/z 39 and m/z 66 transients (see Figure S2). A similar explanation applies to other fragments like m/z 27 and 51 i.e. they also result from a complementary pathway to m/z 66, a conclusion based on their mirrored dynamics and out of phase oscillations with m/z 66. Based on the results of Figure 5, we can conclude that these fragments originate from the broken-bridge intermediate due to their spectral amplitude at 90 cm^{-1} .

10. Same comments (8-9) apply to the other secondary dissociation product assignments in sections 4.2-4.5.

Answer: For better visibility of these transients, a scaled residual line for all ions discussed in the final section of the paper has been added to Figure 4. The data in Figure 4 has been converted to raw data points, the fit used to obtain the residual oscillations is shown as a red line, and a black scaled residual line for all ions has been added above each transient for better phase comparisons. Similar to the mass 39/mass 66 argument above.

We have addressed the same arguments for these other dissociation product assignments directly after the previous paragraph:

As for the product ions originating from the intact (135 cm^{-1} and 195 cm^{-1}) DCPD⁺ intermediate, most share similar dynamics in the pump-probe yields in Figure 4 (m/z 54, 78, 91, 105, 115, 117, and 131) with a long (500 fs) depletion and generally long-lived oscillations. Apart from showing very similar dynamics, these fragments also share an equal branching ratio from the two DCPD⁺ intermediates. In Figure 5, we observe that these fragments show more spectral amplitude at frequencies corresponding to the intact intermediate relative to the other fragments. These similarities in both dynamics and spectral content indicate that fragments in this category may display their dynamics from the same precursor state, most likely the intact DCPD⁺ intermediate. It should be noted that several more transients of this type are shown in the Supplementary Materials (See Figure S1).

Minor comments:

11. In fig. 2, the Asterix indicating the bonds that break is not visible in the figure.

Answer: The asterisks have been added to the figure.

12. On pg. 8, 3 lines from bottom: transient state not necessarily transition-state. I assume it's a typo.

Answer: We understand the confusion because “transition state” has been associated with the saddle point of a reaction since the time of Arrhenius. More recently, the term “transition state” has been used to describe a system that is no longer reagent and not yet product in unimolecular photodissociation by Polanyi. We prefer using the latter to refer to the short-lived species as they evolve into products.

We have added an explanation for the term to avoid confusion, at the end of the first paragraph of the Introduction:

*Here, we augment this capability by mapping coherent vibrational dynamics in the time-dependent product ion yields to short-lived **states of the system that are no longer reagent but not yet product, a term we will now refer to as "transition state" intermediates. Mapping product ions to their originating transition state intermediates helps elucidate mechanistic information about the multiple independent and competing fragmentation pathways taking place following dissociative ionization.***

13. In several places there are typos in reference to figures I guess that appear as “(see ??)” **Answer:** The proper figure references have been added.

14. On pg. 14, please revise the tenses of the "Studies like this shows ..." sentence.

Answer: “Studies like this shows” has been changed to “This study shows”.

Reviewer #2 (Remarks to the Author):

The manuscript “Coherence Mapping to Identify the intermediates of Multi-Channel Dissociative Ionisation” by Stamm et al. details ultrafast pump-probe ion yield measurements on dicyclopentadiene (DCPD), focussing on the retro Diels-Alder reaction following ionisation. In particular the authors singly ionise DCPD with a strong-field pump pulse, and then modulate the fragmentation dynamics with a time-delayed probe pulse. This ‘disruption’ of the fragmentation processes manifests itself as depletion/enhancement of particular fragmentation channels, as well as coherent oscillations in the observed yield. The coherent oscillations in the time-domain are then analysed in terms of their frequency components, which can directly be correlated to the underlying vibrational structure of the intermediates from which a particular fragment originated. This allows the authors to observe a signature for a particular reaction pathway (here stepwise vs. Concerted), and with the aid of ab initial calculations assign each fragment to a particular channel.

The presented approach is a powerful and elegant way to gain insight into the (cation) fragmentation dynamics that are prevalent in MS of larger systems. This is clearly an important development, in particular with a view to predicting complex fragmentation patterns. The chosen process, a retro Diels-Alder reaction, is of high importance to chemistry, while the method of ‘Coherence Mapping’ should be fairly universally applicable. This is an important development and warrants publication in Commun. Chem. That said, the manuscript is currently missing important details that are needed in order to fully understand and reproduce the shown data. The discussion should furthermore be significantly extended, as further outline below. I therefore generally recommend publication after major revisions and further review.

Specifically:

- Most urgently, in Figure 5 the ion MEMs shown at the top (for m/z 54, 66, 131) do not correspond to the data shown in the bottom half. In particular for m/z 66 and 131. My feeling is that the color labels in the top figure are swapped!?

Answer: This is a great catch; the color labels have been swapped. The labels have been switched to match their corresponding ion. See the updated figure below.

• In order to extract vibrational frequencies from the time-resolved data the dynamics component is removed by fitting this with (a sum of) exponential decays representing the lifetimes of intermediate states. I fully agree with this approach, however the resulting fit is not shown in the manuscript (or SI), making it hard to judge how well this described the data. The extracted time-constants are furthermore not given. I strongly recommend that (at least for selected fragments) the actual raw data points, along with the fit and the extracted residuals are shown. The time constants should be discussed and compared to those available in the literature.

Answer: We agree that this is an important part of the work. The data in Figure 4 has been converted to raw data points, the fit used to obtain the residual oscillations is shown as a red line, and a black scaled residual line for all ions has been added above each transient for better phase comparisons. (See the updated figure on Page 4 of this reply.) To the best of our knowledge, literature time constants for the fragmentation of DCPD are not available under the same excitation conditions as shown here. Some of the timescales, however, do match with AIMD simulations. Some discussion regarding this agreement has been added to the Discussion section.

Next to where we show the average timescales obtained via AIMD, we also report the tau's from fitting the experimental m/z 66 transient:

As seen from the AIMD trajectories (See Supplementary Video 1), the formation of CPD via a stepwise mechanism follows the consecutive cleavage of two bridge bonds (labeled with asterisks in Figure 2). From an average of the trajectories, the first bond breaks within 260 fs, and the remaining bridge bond breaks in 2.25 ps. These timescales are reasonably commensurate with timescales obtained by fitting the m/z 66 transient (obtaining timescales of 485 ± 22 fs and 23 ps ± 3 ps). It is interesting to note that the π -electron rearrangement needed for dissociation is modulated by vibrational motion as seen previously in the McLafferty rearrangement. However, in the concerted mechanism, the cleavage of both bridge bonds occurs nearly simultaneously after staying intact for a few picoseconds (See Supplementary Video 2).

• The authors state that “While the Fourier transform is typically used for analysis of such oscillations, we instead use the maximum entropy method (MEM)” - could they comment on why they use MEM instead of the more common FT?

Answer: We agree that this is a choice that should be explained. We chose the MEM as it is more suited for analysis of extremely damped oscillations like those shown in the manuscript.

We added the following sentence before Figure 5:

Fitting the data in Figure 4 to Equation 1 allows us to extract oscillations from the residual, following subtraction of the fit from the data. While the Fourier transform is typically used for analysis of such oscillations, we instead use the maximum entropy method (MEM) as shown in Figure 5. This method was chosen because the MEM is better suited for analyzing the frequencies of short-lived oscillations like those shown in Figure 4. A traditional Fourier transform exhibits a broad peak due to the strong damping of the oscillations. Note that the MEM spectra match the main features of the Fourier transform with much narrower peaks and with some minor peak shifting during retrieval. The multidimensional representation of the data indicates that the frequency at 90 cm^{-1} dominantly modulates most of the fragments. Frequencies 135 cm^{-1} and 195 cm^{-1} are dominant in the larger fragments $m/z \geq 78$ as well as $m/z 54$.

• At the top of page 8 the authors state that “The vibrational frequencies of each fragment in Figure 5 correspond to the vibrational spectrum of the intermediate species from which it originated”. I think this is somewhat overstated. The observed frequencies corresponds to frequencies of the intermediate, they do not represent the entire vibrational spectrum of it.

Answer: We agree that we are only measuring individual vibrational frequencies within the spectrum, and not the entire spectrum.

The sentence has been reworded to emphasize that we are measuring specific frequencies of the spectrum of the intermediates:

The vibrational frequencies of each fragment in Figure 5 correspond to the spectrum of the intermediate species from which it originated, limited by the pulse width and sensitivity of specific vibrational motions to the probing process. The relative phase of the oscillations with respect to each other across different ions can be used to deduce the interplay among different channels. Thus, this information allows us to deduce the reaction mechanism for the dominant fragments, as discussed below.

• For the performed AIMD simulations, how many trajectories were simulated? How were these classed into stepwise or concerted reaction mechanisms?

Answer: We performed 26 trajectories and classified them into stepwise or concerted based on whether both bonds broke within the last 5% of the total reaction time or not. The number of trajectories as well as how they were classified has been added to the AIMD Methods section.

*All trajectories were integrated up to 10 ps using the velocity Verlet integrator with a time step of 0.5 fs. Initial positions and momenta were sampled from the vibrational Wigner distribution using B3LYP/6-31++G** of the neutral ground state of endo-DCPD. All 26 trajectories were visualized using Avogadro and were sorted into concerted or stepwise based on whether both bonds broke within the last 5% of the total reaction time.*

• I feel the discussion could be significantly extended:

Answer: We agree, the entire discussion was rewritten.

• For the discussed correlations it is impossible to see from the small panels in Figure 4 if oscillations are in phase or not. I recommend inserting additional figures that shown in detail the dynamics traces for the particular fragments discussed (e.g. m/z 66 and 39), as well as the extracted residuals showing the oscillatory behaviour.

Answer: This is true, as the comparison between the oscillations are important for the discussion section. A scaled residual line for all ions discussed in the final section of the paper has been added to Figure 4. The data in Figure 4 has been converted to raw data points, the fit used to obtain the residual oscillations is shown as a red line, and a black scaled residual line for all ions has been added above each transient for better phase comparisons. (See the updated figure on Page 4 of this reply.)

- The discussion should moreover be extended to further fragment channels. For example, m/z 79, which shows very different behaviour from m/z 78. Or m/z 115 which is by far the heaviest fragment that appears to follow the step-wise mechanism.

Answer: We agree that this data has lots of information entangled within it. As for discussion about other channels/fragments, an additional section has been added to the discussion about the similarity of several fragment transients.

We have added the following discussion:

As for the product ions originating from the intact (135 cm^{-1} and 195 cm^{-1}) DCPD⁺ intermediate, most share similar dynamics in the pump-probe yields in Figure 4 (m/z 54, 78, 91, 105, 115, 117, and 131) with a long (500 fs) depletion and generally long-lived oscillations. Apart from showing very similar dynamics, these fragments also share an equal branching ratio from the two DCPD⁺ intermediates. In Figure 5, we observe that these fragments show more spectral amplitude at frequencies corresponding to the intact intermediate relative to the other fragments. These similarities in both dynamics and spectral content indicate that fragments in this category may display their dynamics from the same precursor state, most likely the intact DCPD⁺ intermediate. It should be noted that several more transients of this type are shown in the Supplementary Materials (See Figure S1).

- The authors draw the qualitative conclusion that while the rDA process is accessible from both mechanisms, it is dominated by the stepwise reaction. Can they also draw more quantitative conclusions regarding the importance of both channels? Or what would be needed in order for more quantitative insight?

Answer: We report that 90% of the AIMD trajectories show a stepwise mechanism, and the rest a concerted mechanism. To approximate the branching ratio experimentally, we can take the amplitude of the oscillations attributable to each intermediate to get a rough estimate as to which mechanism is more dominant following strong field ionization. This rough calculation for m/z 66 gives an 80:20 ratio for stepwise/concerted mechanism, but it's important to note that this estimate may be affected by different sensitivity to the probe pulse by the two mechanisms. We have added the following comments to the discussion section:

The pump-probe transient of the rDA product (m/z 66) from DCPD⁺ exhibits mostly the low frequency mode at 90 cm^{-1} , which is attributed to the broken-bridge DCPD intermediate. This suggests that the rDA process in DCPD⁺ dominantly occurs via a stepwise mechanism where the DCPD first breaks one of its bridging bonds, undergoes vibrational motion at 90 cm^{-1} in this intermediate state, then breaks the second bridging bond to form the CPD⁺ product (m/z 66). This matches the dominant pathway seen via AIMD simulations. In addition, experimentally we find small contributions from the 135 and 195 cm^{-1} modes in the m/z 66 transient, indicating that a concerted mechanism from the intact molecular ion intermediate is also possible, although not as prevalent as the stepwise mechanism. This is supported by AIMD trajectories, which prevalently show the stepwise mechanism (~90%) as opposed to the concerted mechanism. Experimentally we can estimate the branching ratio between the stepwise and concerted mechanisms by comparing the amplitudes of the frequencies from the two different mechanisms. Doing this for the m/z 66 product ion we obtain a branching ratio of ~80% stepwise and ~20% concerted, in reasonable agreement with the AIMD simulations. Note that this estimate may be influenced by the relative sensitivity of the two different mechanisms to the probe pulse. If one of the pathways is more sensitive to probing than the other, then its measured ratio will be larger due to the more pronounced oscillations.

• The overall structure of the manuscript does not make for easy reading. Especially placing the “Methods” in between the “Results” and “Discussion” section is very unusual. I highly recommend placing this either directly after the “Introduction” or at the end of the manuscript. ex, breaking the “Discussion” into small paragraphs each with their own subheading really breaks the flow of the text. I recommend extending the “Discussion” (as outlined above) and creating a single flowing text.

Answer: We agree with the reviewers’ suggestions. The Methods section has been moved to the end of the manuscript. Additionally, the Discussion section has had its headings removed and instead has transition sentences added between sections for an easier read.

Some minor typos and corrections:

Figure 2 - asterisks highlighting C-C bonds are missing

Answer: The asterisks have been added to the figure.

Figure 6 uses different font and style to the other figures (e.g. round brackets for units instead of square ones)

Answer: The font and style of Figure 6 has been changed to match the rest of the paper. See the updated figure on Page 1 of this reply.

Page 12 - several LaTeX references missing (“See ??”).

Answer: The proper references have been added.

Several references are incomplete: 10, 15, 17, 25, 29, 32

Answer: The missing information in these references has been added.

Reviewer #3 (Remarks to the Author):

The authors present a study of strong-field ionization of the dicyclopentadiene molecule, in which they follow a retro-Diels-Alder reaction of the cation with pump-probe mass spectrometry with a disruptive probe pulse. They observe coherent oscillations in the final fragment ion yields that they attribute to vibrational movement of two distinct reaction intermediates belonging to two different reactions pathways (one concerted, one stepwise).

Such information is highly relevant for the study of cationic fragmentation processes and very challenging to come by with other methods. However, it seems challenging or even impossible with their experimental approach to strictly assign a final product to either of the intermediates, which would be an important second step. The results are novel and relevant for chemists and physicists working in mass spectrometry and photoionization studies. From this point of view, I think the study is suitable for publication in communications chemistry.

However, I have concerns regarding the comparison between theory and experiment and the lack of discussion of parts of the data. Some part of the discussion is difficult to follow because there are no graphs showing the details being discussed. If these concerns can be appropriately addressed, I will recommend publication.

1. Electronic state population of the molecular cation after tunnel ionization

The authors perform a tunnel-ionization experiment, but base their theoretical investigation on the assumption that only the cationic ground state is populated. Is there any argumentation why tunnel ionization should preferentially create ground state cations? Are there references supporting this?

Answer: We agree that tunnel ionization can populate several cationic excited states of DCPD depending on the laser intensity. However, it is reasonable to assume if excited states were populated, rapid relaxation to the cationic ground state would take place, given the dense manifold of cationic states (DCPD has 22 atoms). The rapid relaxation is akin to Kasha's rule in electronic excited states. Within this approximation, the population of excited cation states would manifest primarily as vibrational excitation on the cationic ground state. The total vibrational energy in our simulations (which includes zero-point energy and energy released during adiabatic relaxation) is likely a lower bound to total energy available in the experiment, yet there is more than enough energy to accurately model the fragmentation of $\text{DCPD}^+ \rightarrow \text{CPD}^+ + \text{CPD}$, which is ~ 0.3 eV uphill (see Supplementary Figure 3). We do note that the only reaction that we observed from AIMD trajectories was $\text{DCPD}^+ \rightarrow \text{CPD}^+ + \text{CPD}$, which we used to help explain the experimentally observed frequencies.

To better explain the motivation for running AIMD and acknowledge the limitations in assuming ground state population, the following change to the results and method sections were made:

In the Results section:

To better understand the origin of the experimentally observed coherent oscillations underlying the fragmentation of DCPD, we performed ab initio molecular dynamics calculations (AIMD). Further details of the calculations are presented in the Methods section.

In the Methods section:

*All ab initio molecular dynamics (AIMD) of endo-DCPD were performed on the singly-ionized ground state using uB3LYP/6-31++G** with Wigner initial conditions to approximate experimental conditions, assuming rapid nonadiabatic relaxation within the manifold of cationic states. Trajectories were calculated using the GPU-accelerated quantum chemistry software TeraChem. All trajectories were integrated up to 10 ps using the velocity Verlet integrator with a time step of 0.5 fs.*

In their manuscript the authors state “This indicates that the molecular ion is metastable and preferentially fragments into the prominent peak at m/z 66.” I believe, the mass-selected VUV threshold photoelectron spectrum of dicyclopentadiene (published in [1]) indicates that the DCPD cation is stable in its electronic ground state (without significant vibrational energy). This would indicate that the observed fragment ions are produced from electronic excited states of the cation, populated in the tunnel ionization experiment. Consequently, the comparison to calculations for the cationic ground state would be questionable and would require additional justification. It may be possible that all these electronic states of the cation share the same vibrational signatures depending on whether one of the bridging bonds is broken or not, but this cannot just be assumed without argumentation.

[1.] Pan, Z., A. Puente-Urbina, A. Bodi, et al. Chemical Science, 2021. 12(9)

Answer: While these threshold PES of DCPD indicates that ground state DCPD^+ is stable, we inferred the meta-stability of the DCPD cation formed in our experiments from its low signal at all laser intensities. While simulations were run on ground state DCPD^+ , additional vibrational energy is present following adiabatic relaxation on the cationic ground state. This excess energy (0.8 eV at the B3LYP level), distributed into the reactive modes, is sufficient to drive the reaction. For a unimolecular reaction where $A \rightarrow B + C$, the reaction proceeds because of entropic reasons: B and C cannot find each other to reform A. To better illustrate the reviewer's points in the manuscript the following edits were made to Methods section and Supplementary Figure 3 caption (the figure that shows the potential energy diagram for the reaction $\text{DCPD} \rightarrow \text{CPD}^+ + \text{CPD}$):

In the Main Text:

The experimental mass spectrum of endo-DCPD is shown in Figure 3. It exhibits a small molecular ion peak (m/z 132), even at low intensity conditions. This indicates that, under the excitation and subsequent

internal energy conditions of this experiment, the molecular ion preferentially fragments into the prominent peak at m/z 66.

In the Methods Section:

*All ab initio molecular dynamics (AIMD) of endo-DCPD were performed on the singly-ionized ground state using uB3LYP/6-31++G** with Wigner initial conditions to approximate experimental conditions, assuming rapid nonadiabatic relaxation within the manifold of cationic states. Trajectories were calculated using the GPU-accelerated quantum chemistry software TeraChem. All trajectories were integrated up to 10 ps using the velocity Verlet integrator with a time step of 0.5 fs.*

In the Supplementary Figure 3 Caption:

Despite this reaction being energetically uphill, it can be observed because entropy strongly disfavors the reverse reaction.

2. Why do most products show oscillations with frequencies of both intermediates?

The data in figure 4 shows that most product channels oscillate with all three identified vibrational frequencies. This can either be explained if any of these products is formed via both intermediates, or because the disruptive probe exchanges population between the fragmentation channels (One intermediate is depleted by the disruptive probe toward a product channel of the other intermediate). I think both possibilities need to be discussed. In general, more discussion of the data in the bottom panel of figure 5 would improve the paper.

Answer: This is a very good observation. We believe that both possibilities are reasonable (products arriving from both intermediates or population transfer via the probe). Given the similarities in the DCPD structures following each of the two mechanisms, we find it more probable that each product ion (with all 3 frequencies) is arrived independently from each of the two mechanisms (i.e. each intermediate can lead to several of the product ions). Although the population transfer mechanism is also reasonable. This argument has been added to the discussion section. The following sentences have been added:

*The higher frequency modes lie at 135 cm^{-1} and 195 cm^{-1} . Comparison to theoretically derived frequencies finds close agreement with the two lowest vibrational modes of the singly ionized ground state (calculated to be 150 cm^{-1} and 201 cm^{-1} using uB3LYP/6-31++G**). These modes were also observed in the spectrum obtained from the velocity autocorrelation of AIMD trajectories (146 and 192 cm^{-1} , see Figure 6). These frequencies correspond to internal twisting modes of the intact DCPD and are observed in many of the larger fragment ions ($m/z \geq 78$) as well as m/z 54. The long-lived oscillations imply that the molecular ion can remain intact for several picoseconds, which is also seen in the trajectories that showed a concerted rDA reaction (see Supplementary Video 2). Many ions in Figure 5 show oscillations from both the broken (90 cm^{-1}) and intact (135 and 195 cm^{-1}) intermediates. There are two possible explanations for this observation. The first is that both intermediates can subsequently fragment into almost any ion in the MS (given enough internal energy). The second possibility is that the fragment channels from each intermediate are mostly distinct, and it is the probe pulse that is causing the fragmentation of one intermediate into a channel of the other. We find it more probable that each product ion (with all 3 frequencies) forms independently from each of the two mechanisms (i.e. each intermediate can lead to several of the product ions).*

3. Discussion of relative phases of oscillations

The authors discuss the relative phases of the oscillations and complementary trends observed in different product channels. These phases cannot be observed in the provided graphs. For this discussion it is necessary that the authors provide plots of the residuals showing these oscillations (at the very least in the SI), such that the reader can follow the discussion.

Answer: This is true, as the comparison between the oscillations are important for the discussion section. A scaled residual line for all ions discussed in the final section of the paper has been added to Figure 4. [See updated figure on Page 4 of this reply].

Minor points:

- The legend in figure 5 has masses 66 and 131 interchanged.

Answer: The coloring scheme has been fixed to reflect the data. (See the updated figure on Page 7 of this reply.)

- The time scales given on page 9 do not agree with what I see from the video. The video seems to show bond breaking after 60 fs and after 6 ps rather than 260 fs and 2.25 ps.

Answer: It's important to note that the videos show a single trajectory, while the paper gives time constants that are the average of all available trajectories. This explains the discrepancy in the timescale between the video and the paper.

- In the discussion the CPD⁺ ion is referred to as an intermediate. If I understand correctly, the out-of-phase oscillations of m/z 51 and 39 (not shown!) when compared with m/z 66 shows that these products compete with the m/z 66 (CPD⁺) product. I cannot see any argument why this means that m/z 66 needs to be an intermediate for the production of m/z 51 and 39. This needs to be explained more.

Answer: The reviewer's observation is indeed critical to the interpretation of this experiment.

We have rewritten the explanation as follows:

The second largest peak in the mass spectrum of DCPD corresponds to the cyclopropenium ion (m/z 39), which shows coherent oscillations at 90 cm⁻¹ (Figure 5) similar to CPD⁺ (m/z 66). Comparing the time-resolved curve of m/z 39 with m/z 66 reveals complementary trends in both the timescale and oscillations (see Figure 4). The out-of-phase oscillations and complementary dynamics lead us to conclude that the broken bridge DCPD intermediate has two complementary pathways: one leading to m/z 66 and another leading to m/z 39 (among other fragments). Upon reaching this intermediate, it is prone to fragment into the dominant m/z 66 product, while the probe can excite this intermediate to a state that instead forms m/z 39. Oscillations at the 90 cm⁻¹ frequency modulate the preference for one pathway or another, leading to the out of phase oscillations and mirrored dynamics in the first picosecond of the m/z 39 and m/z 66 transients. It is worth noting that it is possible for m/z 39 to be formed from the fragmentation of m/z 66, as evidenced from the MS of neutral CPD. We see evidence for this conversion, specifically in the long positive time delays (2 ps onwards) of the m/z 39 and m/z 66 transients (see Figure S2). A similar explanation applies to other fragments like m/z 27 and 51 i.e. they also result from a complementary pathway to m/z 66, a conclusion based on their mirrored dynamics and out of phase oscillations with m/z 66. Based on the results of Figure 5, we can conclude that all these fragments originate from the broken-bridge intermediate due to their spectral amplitude at 90 cm⁻¹.

- A comment on the accuracy of the MEM method would be helpful. Are the differences in the 90 cm⁻¹ modes (position as well as widths) observed for the different fragments in figure 5 significant? Could these be indications of different electronic states, with slightly different vibrational frequencies leading to these products?

Answer: The specifics of where the MEM spectrum peaks are inconsequential for the results shown here, as the actual frequency (given by the Fourier transform) is very broad and envelopes all of the shown peaks. Therefore, we don't believe these shifts to mean much on their own. This clarification has been added to the paper.

We added the following sentences before Figure 5:

Fitting the data in Figure 4 to Equation 1 allows us to extract oscillations from the residual, following subtraction of the fit from the data. While the Fourier transform is typically used for analysis of such oscillations, we instead use the maximum entropy method (MEM) as shown in Figure 5. *This method was chosen because the MEM is better suited for analyzing the frequencies of short-lived oscillations like those shown in Figure 4. A traditional Fourier transform exhibits a broad peak due to the strong damping of the oscillations. Note that the MEM spectra match the main features of the Fourier transform with much narrower peaks and with some minor peak shifting during retrieval. The multidimensional representation of the data indicates that the frequency at 90 cm^{-1} dominantly modulates most of the fragments. Frequencies 135 cm^{-1} and 195 cm^{-1} are dominant in the larger fragments $m/z \geq 78$ as well as $m/z 54$.*

- Is the low number of trajectories calculated statistically sufficient? Only two or three trajectories seem to show the concerted decay, is this correct?

Answer: It is true that only a few trajectories show the concerted pathway. We don't claim statistical significance with the AIMD simulations, and the trajectories were run only to gain insights into the different mechanistic details of the rDA reaction following ionization. AIMD was not performed to make quantitative claims about the nature of the mechanisms.

REVIEWERS' COMMENTS:

Reviewer #1 (Remarks to the Author):

The authors addressed my concerns and performed substantial improvements to the manuscript. It is now suitable for publication after a minor correction:

On pg 12 line 9: "It is worth noting that is it possible for... as evidenced..." should it be "it is" ? and "evident" ?

I suggest the authors to carefully scan their manuscript for other typos I may have missed.

Reviewer #2 (Remarks to the Author):

The authors have sufficiently addressed the reviewers comments, and I recommend publication.

Reviewer #3 (Remarks to the Author):

The authors have fully addressed the concerns I had with the original version, and I feel that especially the discussion section has even further been improved. In my original assessment of the importance and novelty of the work nothing has changed. The method of analysing coherent oscillations to identify intermediates and thus pathways of cationic fragmentation reactions is highly relevant and of interest for a wider field of research. Similar experimental information is very difficult to obtain by other means. I therefore recommend publication of the manuscript in communications chemistry.

Reply to Reviewer's Comments:

We are glad that the three reviewers appreciated the improvements made on the manuscript during the past revision. We have corrected a few typos and addressed editorial requests.

Reviewer #1 (Remarks to the Author):

The authors addressed my concerns and performed substantial improvements to the manuscript. It is now suitable for publication after a minor correction:

On pg 12 line 9: "It is worth noting that is it possible for... as evidenced..." should it be "it is" ? and "evident" ?

I suggest the authors to carefully scan their manuscript for other typos I may have missed.

Reviewer #2 (Remarks to the Author):

The authors have sufficiently addressed the reviewers comments, and I recommend publication.

Reviewer #3 (Remarks to the Author):

The authors have fully addressed the concerns I had with the original version, and I feel that especially the discussion section has even further been improved. In my original assessment of the importance and novelty of the work nothing has changed. The method of analysing coherent oscillations to identify intermediates and thus pathways of cationic fragmentation reactions is highly relevant and of interest for a wider field of research. Similar experimental information is very difficult to obtain by other means. I therefore recommend publication of the manuscript in communications chemistry.